# Novel Echocardiographic Measurements of Right Ventricular–Pulmonary Artery Coupling in Predicting the Prognosis of Precapillary Pulmonary Hypertension

**DOI:** 10.3390/jpm13121627

**Published:** 2023-11-21

**Authors:** Weronika Topyła-Putowska, Michał Tomaszewski, Agnieszka Wojtkowska, Andrzej Wysokiński

**Affiliations:** Department of Cardiology, Medical University of Lublin, 20-059 Lublin, Poland; michal.tomaszewski@umlub.pl (M.T.); agnieszka.wojtkowska@umlub.pl (A.W.); andrzej.wysokinski@umlub.pl (A.W.)

**Keywords:** echocardiography, pulmonary hypertension, tricuspid annular plane systolic excursion, pulmonary acceleration time, prognosis

## Abstract

Background: Currently, there are many parameters with proven prognostic significance in pulmonary hypertension (PH). Recently, the parameters defining right ventricular–pulmonary artery coupling (RVPAC) have gained clinical importance. In our study, we investigated the prognostic potential of previously known single echocardiographic parameters and new parameters reflecting RVPAC in patients with precapillary PH. Objective: Our study aimed to evaluate the prognostic value of selected echocardiographic parameters and the neutrophil–lymphocyte ratio (NLR) in adults with precapillary PH. Methods: This study included 39 patients (74% women; average age, 63 years) with precapillary PH: pulmonary arterial hypertension (PAH) and chronic thromboembolic PH (CTEPH). The mean follow-up period was 16.6 ± 13.3 months. Twelve patients (31%) died during the observation time. We measured several echocardiographic parameters, which reflect right ventricular function, pulmonary hemodynamics, and RVPAC. To assess disease progression and the patient’s functional capacity, the World Health Organization functional class (WHO FC) was determined. The patient’s physical capacity was also evaluated using the 6 min walk test (6MWT). The analysis included values of the N-terminal prohormone brain natriuretic peptide (NT-proBNP) and NLR. Results: TAPSE × AcT and TAPSE/sPAP were shown to statistically and significantly correlate with PH predictors, including WHO-FC, 6MWT, and NT-proBNP. Univariate Cox proportional hazards regression analysis revealed that AcT, TAPSE, mPAP, TAPSE/sPAP, RAP, TRPG/AcT, TAPSE × AcT, and NLRs are good predictors of mortality in patients with PH. In addition, the ROC curve analysis showed that TAPSE × AcT is a better predictor of PH-related deaths than TAPSE/sPAP and TAPSE alone. In our study, patients with TAPSE × AcT values < 126.36 had shorter survival times (sensitivity = 72.7%; specificity = 80.0%). Conclusions: TAPSE × AcT is a novel, promising, and practicable echocardiographic parameter reflecting RVPAC, which is comparable to TAPSE/sPAP. Moreover, TAPSE × AcT can be a useful parameter in assessing the severity and prognosis of patients with precapillary PH.

## 1. Introduction

Pulmonary hypertension (PH) is a clinical condition characterized by elevated blood pressure in the pulmonary arteries. PH is either precapillary, postcapillary, or mixed based on its hemodynamic profile. Precapillary PH characteristics include an elevated mean pulmonary artery pressure (mPAP) ≥ 20 mmHg and a normal pulmonary capillary wedge pressure (PCWP) ≤ 15 mmHg, which is measured during right heart catheterization [1]. Precapillary PH develops during conditions in which there is an increase in pulmonary vascular resistance due to various factors, such as changes in the pulmonary vessels or an increase in pulmonary blood flow. This form of PH includes pulmonary arterial hypertension (PAH), PH in the course of lung disease, chronic thromboembolic PH (CTEPH), and PH of an unclear or multifactorial etiology [2].

Both PAH and CTEPH are diseases that are considered rare. The prevalence of these diseases is 48–55 and 3–30 cases per million adults, respectively [3,4]. In the course of PAH, the disease process takes place in the pulmonary vascular bed. The pathophysiology of PAH is based on endothelial dysfunction, which leads to an imbalance between mediators with dilating and constricting effects on the vessels, narrowing the vascular bed [5]. According to registry data, 40–50% of PAH cases are idiopathic PAH (IPAH), 20–30% are PAH in the course of systemic connective tissue diseases, 11–23% are PAH in the course of congenital/corrected heart defects, 10% are PAH in the course of portal hypertension, and 10% are PAH due to drug use [1]. PAH is distinguished from other groups by its aggressive course and poor prognosis [6]. CTEPH develops due to the obliteration of the pulmonary vessel lumen caused by emboli and/or thrombosis. It affects 0.1–9.1% of patients after an acute episode of pulmonary embolism [7].

Due to the debilitating nature of the disease and high mortality, it is important to quickly diagnose, implement effective treatment, and constantly monitor the patient’s condition. The goals of PAH treatment include improving hemodynamic parameters, exercise tolerance, and survival. The most commonly used parameters to assess disease progression include the World Health Organization functional class (WHO FC), hemodynamic parameters assessed during echocardiography, venous blood oxygen saturations, the distances covered in the 6 min walk test (6MWT), and the concentrations of the N-terminal prohormone brain natriuretic peptide (NT-proBNP) [8].

The echocardiographic examination allows for preliminary differential diagnosis and disease progression monitoring. Thanks to this technique, it is possible to assess both the systolic and diastolic dysfunction of the right ventricle, estimate the systolic pressure in the right ventricle, evaluate the acceleration time of flow in the pulmonary artery, tricuspid regurgitation velocity, the collapsibility index of the inferior vena cava, and many other parameters reflecting pulmonary hemodynamics. Techniques for the non-invasive assessment of pulmonary resistance are still being developed and refined, and new indicators are being identified that allow for a prognosis assessment in patients with PH [9,10]. Lately, parameters defining right ventricular–pulmonary artery coupling (RVPAC) have gained clinical importance, assessing the contractile function of the RV and the pressure in the pulmonary artery simultaneously. It seems that parameters defining RVPAC better reflect the progression of this disease and the prognosis in PAH patients than individual echocardiographic parameters [11]. An example of a parameter that reflects RVPAC well and has proven its prognostic significance in PH is TAPSE/sPAP. 

This study aimed to analyze selected laboratory and novel RVPAC echocardiographic parameters and evaluate their prognostic value in patients with precapillary PH.

## 2. Methods Section

The study group consisted of 39 adult patients (74% women) with PAH and CTEPH diagnosed according to the current guidelines [1,12]. The exclusion criteria were as follows: (1) a diagnosis of left heart disease and (2) other forms of PH. The baseline demographic and clinical characteristics of the patients were evaluated. Twelve patients died during the study, and all deaths were due to the exacerbation of right heart failure in the course of the underlying disease.

Patients were enrolled in the study from November 2019 to June 2022, and the end of the observation period was set for 31 December 2022 to obtain at least six months of follow-up. The study was approved by the Bioethics Committee at the Medical University of Lublin, number KE-0254/329/2019.

### 2.1. Echocardiography

All echocardiographic examinations were performed with a Philips EPIQ 7G, Epiq Elite Diagnostic Ultrasound System (Philips Healthcare, Amsterdam, The Netherlands) with 2.5–3.5 MHz transducers by a single investigator (AW). 

In order to assess the basic dimensions of the heart, the linear dimensions of the right and left ventricles were assessed in the four-chamber view (4CV) projection, and then the obtained dimensions were presented in the form of the RV/LV index. In the same projection, the volume of the right atrium was also assessed, striving to obtain the maximum value, as well as the fractional area of change (FAC), which is the quotient of the diastolic and systolic difference in the RV surface area expressed as a percentage. In the parasternal short-axis view, the shape and dimensions of LV and RV were assessed in the end-diastolic and end-systolic phases, and the obtained measurements were presented in the form of the end-systolic and end-diastolic LV eccentricity index.

RV relaxation disorders were assessed by measuring the ratio of the E and A waves of the tricuspid inflow and the measurement of the E wave deceleration time. These values were measured in the 4CV projection on exhalation. Measurements in patients with atrial fibrillation were excluded from the analysis. In addition, using tissue Doppler imaging (TDI), the ratio of early diastolic tricuspid inflow velocity to early diastolic tricuspid annulus velocity medially and laterally (E/e′ ratio) was determined. 

In order to assess the RV systolic function, a tricuspid annular plane systolic excursion (TAPSE), which is the amplitude of the systolic motion of the tricuspid valve annulus, was marked in the M-mode presentation. Using TDI, measurements of the maximal systolic velocity of the tricuspid annulus (S′) were also made. However, while evaluating tricuspid regurgitation with the use of continuous spectral Doppler, the dP/dt parameter was measured, which is an indicator of the rate of pressure increase in the RV contraction phase. 

Right heart hemodynamic measurements included an assessment of right atrial pressure (RAP) based on the size and collapsibility index of the inferior vena cava (IVC) measured in the substernal view. Tricuspid regurgitation velocity (TRV) and tricuspid regurgitation peak gradient (TRPG) measurements were performed using continuous Doppler focused on the regurgitation wave of the tricuspid valve in the 4CV projection. Using TRV and RAP measurements, systolic pulmonary artery pressure (sPAP) was calculated based on the simplified Bernoulli equation. Pulmonary acceleration time (AcT) was measured using the pulsed Doppler, measured in the RV outflow tract. Based on the AcT measurement, mPAP was estimated according to the formula mPAP = 79 − 0.45 × AcT). The parameters TRPG/AcT, TAPSE × AcT, and TAPSE/sPAP were calculated based on the obtained measurements.

### 2.2. Assessment of Disease Progression

In order to assess the stage of disease and prognosis of the patients, the patient’s functional capacity was determined according to the World Health Organization functional class (WHO-FC). Based on the degree of limitation of physical activity and clinical symptoms (dyspnea, fatigue, chest pain, and presyncope), patients were assigned to one of four classes (I–IV). The 6 min walk test (6MWT) (i.e., the distance in meters that a patient is able to walk during a 6 min walk) was also used to assess the patient’s fitness level. Assessments of WHO-FC and 6MWT were performed on the day of echocardiography. The values of the NLR and biochemical parameter NT-proBNP were also used to assess the prognosis, and they were measured on the day of echocardiography in most cases. If it was not possible to perform the measurement on the day of the echocardiographic examination, the NT-proBNP value measured during the next follow-up visit/hospitalization was used for analysis. All NT-proBNP values included in the analysis were marked +/−40 days from the date of echocardiography.

## 3. Statistical Analysis

Statistical analysis was performed using GraphPad Prism 9 (GraphPad Software, San Diego, CA, USA). Data expressed on a quantitative scale were presented as the mean and SD. Data expressed on a qualitative scale were presented as the number and percentage of the sample. Depending on the result of the Shapiro–Wilk test (assessment of compliance with the normal distribution), Pearson or Spearman correlation analysis was applied. Additionally, Student’s *t*-test, the Mann–Whitney U test, one-way ANOVA with Tukey’s post hoc test, and the Kruskal–Wallis test with Dunn’s post hoc test were used. In addition, receiver operating characteristic (ROC) curves were analyzed to obtain the cut-off values of the tested variables that best-distinguished patient survival. The Kaplan–Meier method was employed to plot survival curves. A log-rank test was also used to compare the survival curves with the Cox proportional hazards test. Results were considered statistically significant when *p* < 0.05.

## 4. Results

### 4.1. General Results

The study group included 39 patients (29 women and 10 men) with precapillary PH diagnosed according to current guidelines [1]. There were 33 patients with PAH (idiopathic PAH, *n* = 12; connective tissue disease PAH, *n* = 7; PAH associated with congenital defects, *n* = 13; portopulmonary hypertension, *n* = 1); and six patients with CTEPH. The mean age of the patients was 63.1 ± 15.9 years. The mean follow-up time was 16.6 ± 13.3 months, during which 12 patients died, which is 31% of the study group. 

All patients were on a PAH- or CTEPH-specific treatment during the study. The general clinical characteristics of the study group are presented in Table 1. 

Detailed characteristics of the study group divided into survivors and non-survivors are shown in Table 2. Non-survivors were characterized by lower 6MWT values, higher NT-proBNP and NLR values, and a shorter survival time. The vast majority of non-survivors (83.33%) were in WHO class IV compared to survivors, where only 29.63% of patients were in WHO class IV. Among the echocardiographic parameters, non-survivors were characterized by statistically significant lower values of AcT, TAPSE, TAPSE × AcT, and TAPSE/sPAP and statistically significant higher values of TRPG/AcT, RAP, and mPAP. 

### 4.2. Echocardiography

All echocardiographic parameters were compared with the disease prognostic markers, including WHO-FC, 6MWT, and NT-proBNP. We observed a significant positive correlation between TAPSE × AcT and the results of the 6MWT (r = 0.66; *p* = 0.0001) and a negative correlation between TAPSE × AcT and NT-proBNP values (r = -0.48; *p* = 0.0079), as shown in Figure 1.

Statistically significant higher TAPSE × AcT values were found in the WHO-FC II group compared to the WHO-FC IV group (median 172.660 vs. 108.800; *p* = 0.0324), as well as the WHO-FC III group compared to the WHO-FC IV group (median 184.800 vs. 108.800; *p* = 0.0182). There were no statistically significant differences between the WHO-FC II and III groups (Figure 2).

Furthermore, our study revealed that TAPSE/sPAP correlated positively with the results of the 6MWT (r = 0.39; *p* = 0.0240) and negatively correlated with NT-proBNP values (r = 0.42; *p* = 0.0238 (Figure 3). 

Additionally, our analysis found significantly higher TAPSE/sPAP values in the WHO-FC II group than in the WHO-FC IV group (median 0.314 vs. 0.200; *p* = 0.0279). There were no statistically significant differences between the other groups (Figure 4.).

Univariate Cox proportional hazards regression analysis revealed a few echocardiographic predictors of mortality in patients with PH, as shown in Table 3.

Receiver operating characteristic (ROC) investigations determined that TAPSE, TAPSE/sPAP, TAPSE × AcT, and TRPG/AcT are good predictors of mortality in PH patients (All AUCs presented in Table 4). Additionally, ROC analysis revealed a higher AUC for TAPSE × AcT than for TAPSE and TAPSE/sPAP alone (AUC 0.777 vs. AUC 0.756 vs. AUC 0.746). 

The ROC curve for the TAPSE variable had an AUC of 0.756 (*p* = 0.011). The optimal cut-off point for predicting mortality was set at 16.50 mm (sensitivity 75.0%; specificity 70.4%). The ROC curve for the TAPSE/sPAP variable had an AUC of 0.746 (*p* = 0.0039), and the cut-off point was set at 0.242 mm/mmHg (sensitivity 90.0%; specificity 54.2%). The ROC curve for the TAPSE × AcT variable had an AUC of 0.777 (*p* = 0.0028), and the optimal cut-off value was set at 126.36 mm * ms (sensitivity 72.7%; specificity 80.0%). Additionally, ROC analysis confirmed the role of TRPG/AcT in predicting mortality (AUC 0.727, *p* = 0.0138) and revealed a cut-off value of 0.91 mmHg:m/s (sensitivity 63.6%; specificity 65.0%) (Figure 5 and Figure 6). Among the analyzed parameters, TAPSE/sPAP had the highest sensitivity (90.0%), while specificity was the highest for TAPSE × AcT (0.800).

## 5. Discussion

Echocardiography is often the first study that raises the suspicion of PH. As a non-invasive, widely available, and cheap test, it is an ideal method that can be used for initial diagnosis and should be performed in every patient with suspected PH in order to assess its probability. Moreover, in clinical practice, it is helpful when monitoring the clinical condition and estimating the prognosis, as many echocardiographic parameters have a documented prognostic value in PH [13,14,15]. A well-known parameter with a proven prognostic value is TAPSE [16]. This parameter reflects the function of the RV longitudinal fibers and is an indicator of the RV’s systolic function [17]. A TAPSE < 18 mm is associated with a poorer prognosis and higher mortality rate in patients with PH [18]. In addition, there are several parameters that accurately reflect the function of the right heart and can indirectly be used to estimate the pressure in the pulmonary artery [19]. In clinical practice, sPAP is most often calculated based on the Bernoulli equation, using the TRV and RAP values [20]. Over the years, the role of TRV has grown in value in the context of evaluating patients with PH. According to the latest guidelines, TRV can be used as an independent parameter for the echocardiographic estimation of the probability of PH [6]. 

Recently, the assessment of RVPAC (the simultaneous assessment of RV systolic function and pulmonary artery pressure) has gained clinical importance [21]. Numerous studies have shown that RV contractility, in combination with hemodynamic indices, may provide a more precise assessment of cardiovascular capacity than single parameters [22,23,24]. An example of a parameter that reflects RVPAC well and that has a proven prognostic value in PH is TAPSE/sPAP [25,26]. It has been proven that this parameter is associated with hemodynamics and functional class in patients with PAH [27]. Patients with TAPSE/sPAP < 0.31 mm/mmHg had a significantly worse prognosis than those with higher values [28]. According to our previous analysis, another parameter that reflects pulmonary hemodynamics well is TRV/TAPSE. TRV/TAPSE correlates with prognostic markers of PH, such as WHO-FC, the 6MWT, NT-proBNP levels, and survival in adults with PH [29]. TAPSE × AcT seems to be a new parameter that adequately reflects RVPAC. This parameter has not yet been described in the PH patient population, but in the heart failure patient population, TAPSE × AcT has been shown to be independently associated with mortality, resulting in it being the second-best predictive measure of RVPAC following the TAPSE/sPAP ratio [30]. Another new parameter reflecting the idea of RVPAC is TRPG/AcT. It has been proven that values > 0.6 identify patients with an elevated risk of CTEPH after pulmonary embolism [31]. This parameter was tested in a group of PAH patients for the first time in our study. 

A comprehensive diagnostic approach to patients should also include an assessment of clinical, biochemical, and hemodynamic parameters. The World Health Organization’s functional classification, WHO FC, which is a New York Heart Association (NYHA) classification adapted for use in patients with pulmonary hypertension, is used for clinical evaluation in PH patients. Patients at a high clinical stage (class IV according to WHO-FC) have the worst prognosis [32,33]. The WHO-FC assessment can be extended with methods that allow for a more measurable evaluation of the patient’s fitness level. The 6MWT is especially useful, with a distance <400 m in the 6MWT as an unfavorable prognostic indicator [34]. Using biochemical parameters is another non-invasive monitoring method that provides a good representation of right ventricular function and overload in patients with PH. In clinical practice, the determinations of natriuretic peptides (BNP, NT-proBNP) are used, from which an elevated value should be treated as a negative prognostic parameter [35]. Recent reports indicate that NLR plays a role as an independent predictor of event-free survival in PAH [36,37]. Our study confirmed that NLR is a good predictor of mortality in patients with precapillary PH (HR 1.07, *p* = 0.005).

Our statistical analysis identified several echocardiographic parameters and one laboratory parameter that could be clinically applicable in assessing the prognosis and stratification of patients with PH. TAPSE × AcT and TAPSE/sPAP were shown to significantly correlate with PH predictors, including WHO-FC, 6MWT, and NT-proBNP. Both the values of TAPSE × AcT and TAPSE/sPAP decrease with increasing NT-proBNP values and with deterioration in the WHO-FC of PH patients. In addition, patients with lower TAPSE × AcT and TAPSE/sPAP values performed worse in the 6MWT. Therefore, the TAPSE × AcT and TAPSE/sPAP parameters may be indicators of the severity of the disease and may be helpful in the risk stratification of patients with PH. These correlations have already been demonstrated in previous studies in the case of TAPSE/sPAP in patients with PAH [38,39]. However, our study is the first to prove the mentioned correlations with TAPSE × AcT in a group of patients with precapillary PH. In addition, ROC curve analysis showed that TAPSE × AcT is a better predictor of PH-related death than TAPSE/sPAP and TAPSE alone (AUC 0.777 vs. AUC 0.746 vs. AUC 0.756). TAPSE × AcT may be an alternative parameter for RVPAC assessment when sPAP assessment is not possible (e.g., due to limited IVC visualization) [40]. Moreover, there are reports that estimating RAP based on the diameter and collapsibility of the IVC gives imprecise results [41,42]. On the other hand, it is worth remembering that cardiac function and the presence of tricuspid regurgitation do not affect AcT changes, while both a decrease in the cardiac index and an increase in pulmonary blood flow are associated with left/right shunts “above” the tricuspid valve, which prolongs AcT [43,44].

Univariate Cox proportional hazards regression analysis revealed that AcT, TAPSE, mPAP, TAPSE/sPAP, RAP, TRPG/AcT, and TAPSE × AcT, and NLR are good predictors of mortality in patients with PH. Focusing on the parameters for evaluating RVPAC, we assessed the Kaplan–Meier survival probabilities of TAPSE/sPAP, TRPG/AcT, TAPSE × AcT for comparison with a well-known prognostic parameter, TAPSE. The risk of death was higher for patients with lower values of TAPSE, TAPSE/sPAP, and TAPSE × AcT. However, the probability of survival for the value of 0.91 for the TRPG/AcT parameter turned out to be statistically insignificant. In the Kaplan–Meier probability of survival, patients with TAPSE/sPAP values < 0.242 had poorer survival (*p* = 0.0039) (sensitivity = 90%; specificity = 54.2%). Kaźmierczyk et al., in their study, reported a similar value of TAPSE/sPAP, 0.25, which indicated a particularly poor prognosis in the group of patients with PAH [45]. In our study, patients with TAPSE × AcT values < 126.36 had shorter survival times (sensitivity = 72.7%; specificity = 80.0%). By contrast, in the case of TAPSE, the cut-off value was 16.00 mm (sensitivity = 75.0%; specificity = 70.4%).

In our study, we distinguished two echocardiographic parameters that deserve special attention in the context of assessing the prognosis of patients with PH. We proved that there is a correlation between TAPSE/sPAP and TAPSE × AcT with disease severity markers, such as WHO-FC, 6MWT, and NT-proBNP, and survival in a group of adult patients with PAH and CTEPH. Based on these results, it can be concluded that these parameters may be useful in the assessment of risk stratification and may identify patients with poor prognoses. In addition, TAPSE × AcT was tested for the first time in a group of patients with precapillary PH, and it seems to be a promising parameter for assessing RVPAC in this group of patients. It is worth noting that TAPSE × AcT is measurable using the standard M-mode and Doppler echocardiography; therefore, it could be readily available for practical clinical use.

## 6. Limitations

The main limitation of this study is the small study group. The indicated parameters, especially the new TAPSE × AcT parameter, should be tested on a larger study group of patients in order to confirm the clinical usefulness of this parameter. The value of this parameter, which is associated with a worse prognosis, should be validated in multicenter studies on a larger group of patients. In addition, there were no patients in WHO class I in the study group; therefore, our study did not test the clinical utility of this parameter in a group of patients with mild disease. One of the major limitations of this study is the lack of data from right heart catheterization and a comparative analysis with the echocardiographic parameters included in this paper. Hemodynamic data were not considered in this study due to the significant time discrepancy between the right heart catheterization and echocardiography.

## 7. Conclusions

TAPSE × AcT is a promising, newly proposed echocardiographic parameter reflecting RVPAC. In our study, TAPSE × AcT and TAPSE/sPAP strongly correlated with the prognostic markers of PH, including WHO-FC, the 6MWT, and NT-proBNP levels, and survival in patients with precapillary PH. They may be useful for the stepwise echocardiographic risk stratification of PH patients, with the ability to identify patients with a poor prognosis. Additionally, NLR may be a valid laboratory indicator of mortality in patients with precapillary PH.

## Figures and Tables

**Figure 1 jpm-13-01627-f001:**
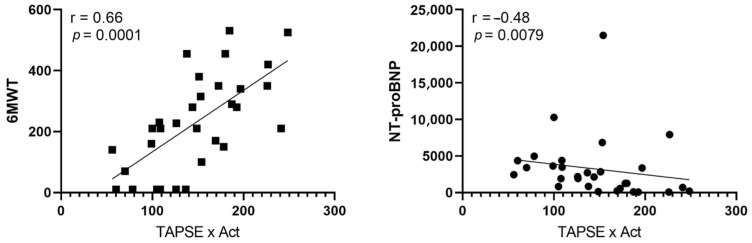
Pearson correlation between 6MWT and TAPSE × AcT (r = 0.66; *p* = 0.0001) and Spearman correlation between NT-proBNP and TAPSE × AcT (r = -0.48; *p* = 0.0079). 6MWT, 6 min walk test; NT-proBNP, N-terminal pro-B-type natriuretic peptide; TAPSE × AcT tricuspid annular plane systolic excursion × pulmonary acceleration time product.

**Figure 2 jpm-13-01627-f002:**
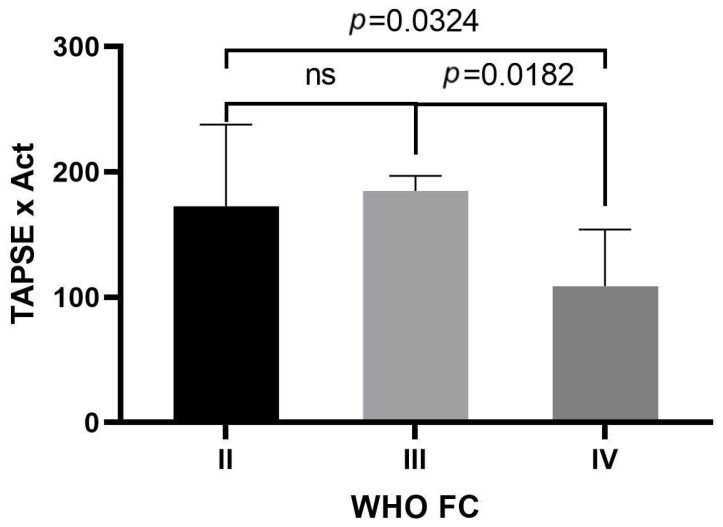
TAPSE × AcT levels in groups of patients with WHO II, III, and IV classes. One-way ANOVA with Tukey’s post hoc test. WHO-FC, World Health Organization functional class; TAPSE × AcT tricuspid annular plane systolic excursion × pulmonary acceleration time product.

**Figure 3 jpm-13-01627-f003:**
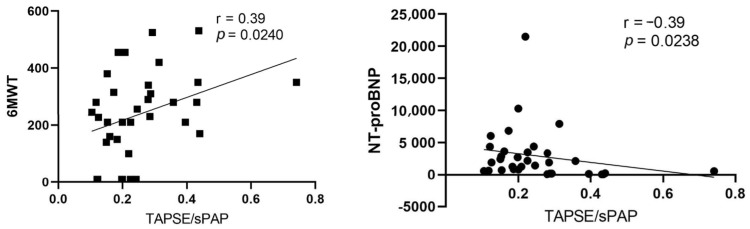
Spearman correlations between 6MWT and TAPSE/sPAP (r = 0.39; *p* = 0.0240) and between NT-proBNP and TAPSE/sPAP (r = −0.39; *p* = 0.0238). 6MWT, 6 min walk test; NT-proBNP, N-terminal pro-B-type natriuretic peptide; TAPSE/sPAP, tricuspid annular plane systolic excursion/systolic pulmonary artery pressure ratio.

**Figure 4 jpm-13-01627-f004:**
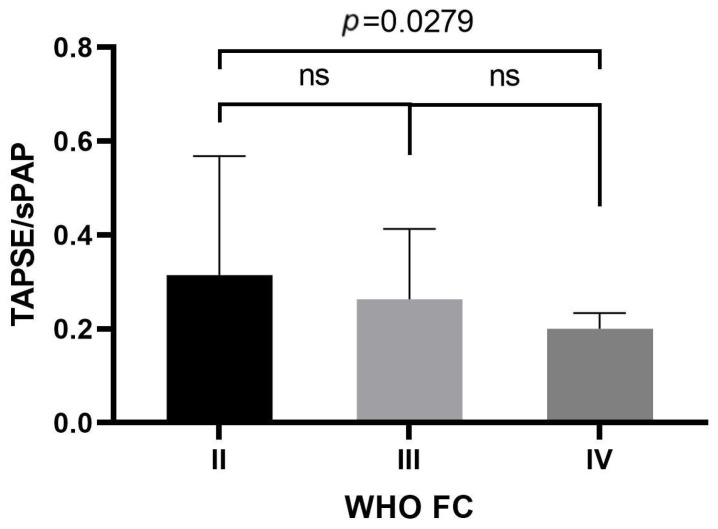
TAPSE/sPAP levels in the groups of patients with WHO-FC II, III, or IV class according to the Kruskal–Wallis test with Dunn’s post hoc test. WHO-FC, World Health Organization functional class; TAPSE/sPAP, tricuspid annular plane systolic excursion/systolic pulmonary artery pressure ratio.

**Figure 5 jpm-13-01627-f005:**
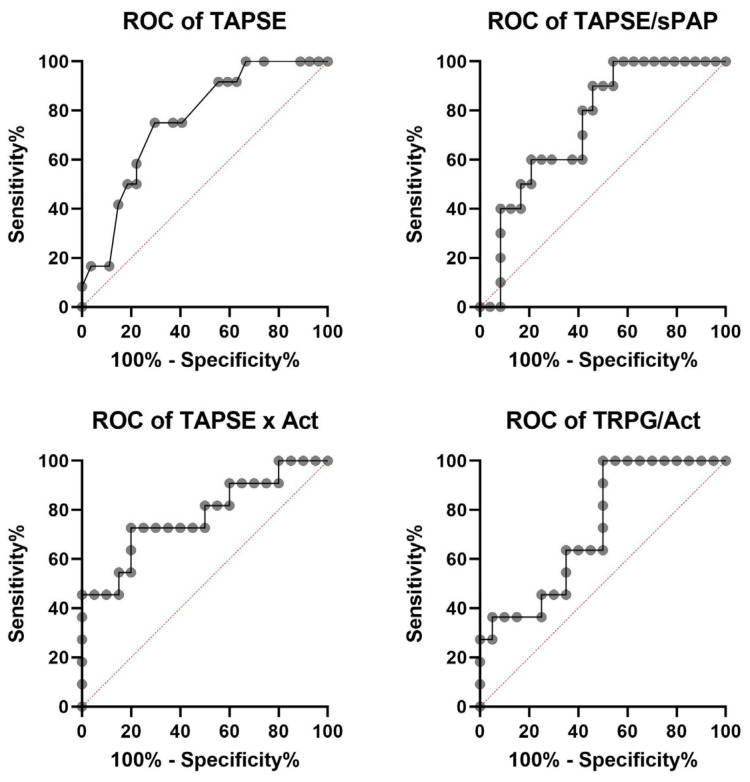
Receiver operating characteristic (ROC) analysis of TAPSE, TAPSE/sPAP, TAPSE × AcT, and TRPG/AcT for the endpoint of all-cause mortality in 39 patients with precapillary PH (all deaths n = 12). TAPSE, tricuspid annular plane systolic excursion; TAPSE/sPAP, tricuspid annular plane systolic excursion/systolic pulmonary artery pressure ratio; TAPSE × AcT, tricuspid annular plane systolic excursion × pulmonary acceleration time product; TRPG/AcT ratio, tricuspid regurgitation peak gradient/pulmonary acceleration time ratio.

**Figure 6 jpm-13-01627-f006:**
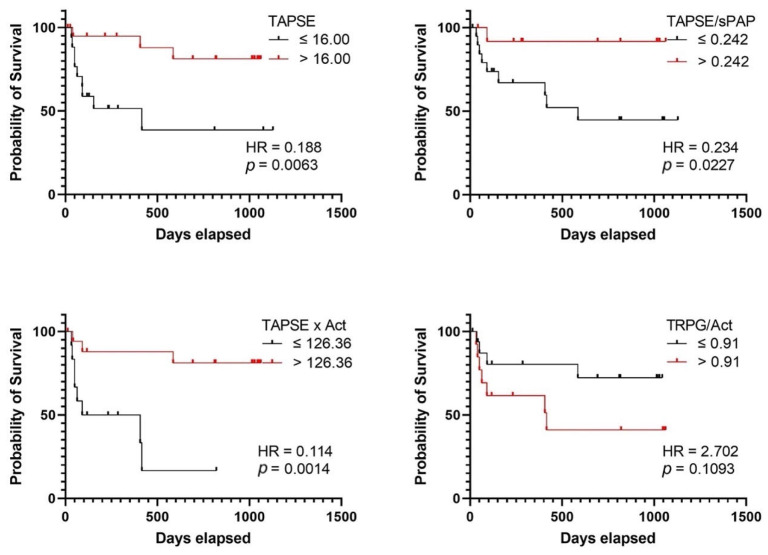
Kaplan–Meier curves presenting impoverished-free survival in precapillary PH. TAPSE, tricuspid annular plane systolic excursion; TAPSE/sPAP, tricuspid annular plane systolic excursion/systolic pulmonary artery pressure ratio; TAPSE × AcT, tricuspid annular plane systolic excursion × pulmonary acceleration time product; TRPG/AcT ratio, tricuspid regurgitation peak gradient/pulmonary acceleration time ratio.

**Table 1 jpm-13-01627-t001:** General characteristics of the study patients.

General Characteristic	
Age, years	63.1 ± 15.9
Female gender, % (*n*)	74% (29)
BMI, kg/m^2^	23.5 ± 3.1
PH etiology	
IPAH, % (*n*)	30.8% (12)
CTD-PAH, % (*n*)	33.3% (13)
CHD-PAH, % (*n*)	17.9% (7)
PoPH, % (*n*)	2.6% (1)
CTEPH, % (*n*)	15.4% (6)
Comorbidities	
Hypertension, % (*n*)	66.7% (26)
Diabetes, % (*n*)	41.0% (16)
Obesity, % (*n*)	23.1% (9)
Hyperlipidemia, % (*n*)	56.4% (22)
Chronic kidney disease, % (*n*)	17.9% (7)
Heart failure, % (*n*)	41.0% (16)
Atrial fibrillation, % (*n*)	43.6% (17)
Ischemic heart disease, % (*n*)	28.2% (11)
Chronic obstructive pulmonary disease, % (*n*)	12.8% (5)
PH treatment	
Endothelin receptor antagonist, % (*n*)	61.5% (24)
Phosphodiesterase-5 inhibitors, % (*n*)	82.1% (32)
Prostanoids, % (*n*)	43.6% (17)
Stimulator of soluble guanylate cyclase, % (*n*)	7.7% (3)
Agonists of the prostacyclin receptor, % (*n*)	15.4% (6)
Diuretics, % (*n*)	43.6% (17)

IPAH, idiopathic PAH; CTD-PAH, connective tissue disease PAH; CHD-PAH, congenital heart disease PAH; PoPH, portopulmonary hypertension; CTEPH, chronic thromboembolic PH.

**Table 2 jpm-13-01627-t002:** Summary presentation of the echocardiographic parameters of the study group.

	Survivors	Non-Survivors	*p*-Value
Women. % (*n*)	72.41% (21)	27.59% (8)	0.6927 ^&^
Men. % (*n*)	60% (6)	40% (4)	
Age. y	58.48 ± 17.05	68.00 ± 12.25	0.1360 ^#^
Neutrophil-to-lymphocyte ratio	3.34 ± 2.01	7.99 ± 10.23	0.0668 ^#^
Mean survival time. days *	593.78 ± 430.69	168.08 ± 189.23	0.0065 ^#^
WHO FC II	22.22%	0.00%	0.0030 ^&^
WHO FC III	48.15%	16.67%	
WHO FC IV	29.63%	83.33%	
Echocardiographic parameters			
FAC. %	0.34 ± 0.12	0.37 ± 0.16	0.6025 ^
RV dp/dt. mmHg/s	874.66 ± 424.16	857.91 ± 327.48	0.9266 ^
S′. cm/s	11.82 ± 2.37	10.73 ± 1.49	0.2712 ^
AcT. m/s	86.35 ± 17.16	74.27 ± 21.78	0.0287 ^#^
TRPG. mmHg	0.81 ± 0.35	1.12 ± 0.41	0.2915 ^
TRV. m/s	4.14 ± 0.78	4.43 ± 0.53	0.2500 ^
TRPG/AcT. mmHg:m/s	0.81 ± 0.35	1.12 ± 0.41	0.0300 ^
E/A ratio	1.08 ± 0.56	1.24 ± 0.40	0.3500 ^#^
Lateral E/e’ratio	10.39 ± 6.00	11.4 ± 8.82	0.6958 ^#^
Medial E/e’ratio	5.83 ± 2.70	6.45 ± 1.81	0.2895 ^
E wave deceleration time. ms	233.49 ± 116.01	180.4 ± 96.57	0.3024 ^
TAPSE. mm	18.48 ± 3.90	14.96 ± 3.15	0.0094 ^
TAPSE × AcT. mm × s	166.15 ± 45.33	113.99 ± 47.52	0.0053 ^
End-systolic eccentricity index	1.47 ± 1.17	1.66 ± 0.55	0.1336 ^#^
Diastolic eccentricity index	1.22 ± 0.35	1.56 ± 0.57	0.0742 ^#^
RAA. cm^2^	24.83 ± 6.53	29.95 ± 7.93	0.0524 ^
RAP. mmHg	6.61 ± 3.35	13.71 ± 3.45	0.0016 ^#^
RV/LV ratio	1.16 ± 0.36	1.33 ± 0.36	0.1733 ^
mPAP. mmHg	39.50 ± 7.87	47.61 ± 7.83	0.0182 ^#^
sPAP. mmHg	75.20 ± 25.85	88.90 ± 20.49	0.1448 ^
TAPSE/sPAP. mm/mmHg	0.29 ± 0.14	0.18 ± 0.06	0.0270 ^#^

* Number of days from the performance of the echocardiogram to the end of the observation time; #—Mann–Whitney test; ^—Student’s *t*-test; &—chi^2^ test. 6MWT, 6 min walk test; NT-proBNP, N-terminal pro-B-type natriuretic peptide; WHO-FC, World Health Organization functional class; FAC, fractional area change; RV, right ventricle; AcT, pulmonary artery acceleration time; TRPG, tricuspid regurgitation peak gradient; TAPSE, tricuspid annular plane systolic excursion; TRV, tricuspid regurgitation velocity; RAA, right atrial area; RAP, right atrial pressure; RV/LV, right ventricular to left ventricular diameter ratio; mPAP, mean pulmonary arterial pressure; sPAP, systolic pulmonary artery pressure.

**Table 3 jpm-13-01627-t003:** Univariate Cox regression analysis for all-cause mortality.

	Mortality	
	HR	*p*-Value
AcT. m/s	0.96 (0.92–0.99)	0.05
TAPSE. mm	0.82 (0.70–0.95)	0.007
mPAP. mmHg	1.13 (1.02–1.24)	0.02
TAPSE/sPAP. mm/mmHg	0.0001 (0.00–0.87)	0.005
RAP mmHg	1.25 (1.07–1.45)	0.005
TRPG/AcT ratio. mmHg:m/s	5.13 (1.19–22.15)	0.03
TAPSE × Act. mm × s	0.98 (0.96–0.99)	0.03
Neutrophil-to-lymphocyte ratio	1.07 (1.03–1.70)	0.005

AcT, acceleration time; TAPSE, tricuspid annular plane systolic excursion; mPAP, mean pulmonary artery pressure; TAPSE/sPAP, tricuspid annular plane systolic excursion/systolic pulmonary artery pressure ratio; RAP, right atrial pressure; TRPG/AcT ratio; tricuspid regurgitation peak gradient/pulmonary acceleration time ratio; TAPSE × AcT, tricuspid annular plane systolic excursion × pulmonary acceleration time product.

**Table 4 jpm-13-01627-t004:** Receiving operating characteristic analysis for all-cause mortality.

	Area under the Curve	*p*-Value	Best Cut-off Value	Sensitivity	Specificity
TAPSE	0.756 (0.602–0.911)	0.0011	16.50 mm	0.750	0.704
TAPSE/sPAP	0.746 (0.579–0.913)	0.0039	0.242 mm/mmHg	0.900	0.542
TAPSE × AcT	0.777 (0.595–0.959)	0.0028	126.36 mm × s	0.727	0.800
TRPG/AcT	0.727 (0.546–0.908)	0.0138	0.91 mmHg:m/s	0.636	0.650

TAPSE, tricuspid annular plane systolic excursion; TAPSE/sPAP, tricuspid annular plane systolic excursion/systolic pulmonary artery pressure ratio; TAPSE × AcT, tricuspid annular plane systolic excursion × pulmonary acceleration time product; TRPG/AcT ratio, tricuspid regurgitation peak gradient/pulmonary acceleration time ratio.

## Data Availability

The data are contained within the article or are available from the authors upon reasonable request.

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
