# Peer review of "Novel Echocardiographic Measurements of Right Ventricular–Pulmonary Artery Coupling in Predicting the Prognosis of Precapillary Pulmonary Hypertension"

_jpm, 2023, doi:10.3390/jpm13121627_

Round 1

Reviewer 1 Report

Comments and Suggestions for Authors

In this interesting manuscript the authors examine some echocardiographic parameters of RV-PA coupling and one laboratory parameter (NLR) as prognostic parameters in precapillary PH.

Minor comments

line 33: precapillary, postcapillary or mixed

Table 2: age and sex should also be compared between survivors-non-survivors

lines 218-225 and 329-333: correlation between TAPSE/sPAP and TAPSE x AcT is conceptually not only expected, but almost sure, as TAPSE is common in both parameters and sPAP and AcT both reflect pulmonary pressure. In my opinion this analysis should be removed (not included), as it does not add significant information.

The lack of hemodynamic data and the inability to perform correlation/regression analysis between the selected parameters and hemodynamic ones is a major limitation of the study and this should be stated.

What about NLR? Does it correlate with parameters of RV-PA coupling or pulmonary pressure or RV conractility? Does it correlate with NTproBNP?

Author Response

Dear Reviewer,

Point 1:
line 33: precapillary, postcapillary or mixed

Response 1:
Following the Reviewer’s suggestion, we corrected indicated sentence in the manuscript.

Point 2:
Table 2: age and sex should also be compared between survivors-non-survivors

Response 2:
As proposed by the Reviewer, we added the indicated data to Table 2.

Point 3:
lines 218-225 and 329-333: correlation between TAPSE/sPAP and TAPSE x AcT is conceptually not only expected, but almost sure, as TAPSE is common in both parameters and sPAP and AcT both reflect pulmonary pressure. In my opinion this analysis should be removed (not included), as it does not add significant information.

Response 3:
As suggested by the Reviewer, we removed indicated analysis from the paper.

 Point 4:         
The lack of hemodynamic data and the inability to perform correlation/regression analysis between the selected parameters and hemodynamic ones is a major limitation of the study and this should be stated.

Response 4:
We agree with the Reviewer and, as proposed, we have highlighted in the sixth paragraph that the lack of right heart catheterization data is an important limitation of the study.

Point 5:         
What about NLR? Does it correlate with parameters of RV-PA coupling or pulmonary pressure or RV conractility? Does it correlate with NTproBNP?

Response 5:
Dear Reviewer,         
in our study, NLR were compared with and all echocardiographic parameters included in the paper. Statistically significant correlation was shown only between NLR and fractional area change (FAC), p-value= 0.048. There were no statistically significant correlation between NLR and pulmonary pressure, and other RVPAC parameters.     

Reviewer 2 Report

Comments and Suggestions for Authors

This study demonstrates for the first time that TAPSE x AcT is an echocardiographic index that reflects the prognosis of precapillary pulmonary hypertension. This study was well designed, statistically detailed, and the results were well summarized. 

This reviewer has the following comments:

1) The study included a relatively large number of patients with hypertension, diabetes, dyslipidemia, atrial fibrillation, and ischemic heart disease, and we are concerned about the effects of left ventricular dysfunction, especially left ventricular diastolic dysfunction.

2) TAPSE/sPAP is physiologically meaningful as an index of right ventricular-pulmonary artery coupling, but can TAPSE x AcT and TAPSE/AcT be physiologically meaningful? Can both parameters be used as indicators of right ventricular pulmonary artery coupling?

Author Response

Dear Reviewer,

Point 1:         
The study included a relatively large number of patients with hypertension, diabetes, dyslipidemia, atrial fibrillation, and ischemic heart disease, and we are concerned about the effects of left ventricular dysfunction, especially left ventricular diastolic dysfunction.

Response 1:
Dear Reviewer,

in the echocardiograms we performed, several parameters were evaluated, mainly focusing on right ventricular function. Regarding left ventricular function, we assessed LV systolic function (LV ejection fraction), mean value was 58.68 % ± 6.52, which is within the range of normal values. This parameter was not included in the paper, as the study focused on right ventricular function and pulmonary hemodynamics. Unfortunately, parameters reflecting LV diastolic function were not assessed in our study.  As suggested by the Reviewer, we will also include the mentioned parameters in future studies.

Point 2:
TAPSE/sPAP is physiologically meaningful as an index of right ventricular-pulmonary artery coupling, but can TAPSE x AcT and TAPSE/AcT be physiologically meaningful? Can both parameters be used as indicators of right ventricular pulmonary artery coupling?

Response 2:
Dear Reviewer,

AcT is a parameter reflecting the mean pulmonary artery pressure (mPAP) and their values are inversely proportional to each other. Thus, it can be concluded that AcT correlates negatively with RV afterload.  TAPSE, meanwhile, is a marker of longitudinal systolic RV function. Both parameters in combination represent the right ventricular-pulmonary artery coupling (RVPAC), the relationship between RV contractility and RV afterload, similar to TAPSE/sPAP. TAPSE x AcT is conceptually different from TAPSE/sPAP, because RV function is collated with mean PAP rather than systolic PAP. Quoting Chemla et al. " the mean PAP reflects the steady component of the circuit and the functional status of the distal (resistive) pulmonary vasculature, while SPAP relates, for a given mean PAP, to the pulsatile component of the circuit, which includes the characteristics of RV ejection and those of the proximal (elastic) pulmonary arteries and wave reflections." [1]. Our study further showed that TAPSE x AcT significantly correlates with TAPSE/sPAP.